# Using RGISTools to Estimate Water Levels in Reservoirs and Lakes

**Ana F. Militino** [1,2,3,*], **Manuel Montesino-SanMartin** [1,2], **Unai Pérez-Goya** [1,2,3] and **M. Dolores Ugarte** [1,2,3]

[1] Department of Statistics, Computer Science and Mathematics, Public University of Navarre,
31006 Pamplona, Spain; manuel.montesino@unavarra.es (M.M.-S.); unai.perez@unavarra.es (U.P.-G.);
lola@unavarra.es (M.D.U.)

[2] Institute for Advanced Materials and Mathematics (InaMat²), Public University of Navarre,
31006 Pamplona, Spain

[3] Department of Mathematics, UNED Pamplona, 31006 Pamplona, Spain

\* Correspondence: militino@unavarra.es; Tel.: +34-948-169206

**Abstract:** The combination of freely accessible satellite imagery from multiple programs improves the spatio-temporal coverage of remote sensing data, but it exhibits barriers regarding the variety of web services, file formats, and data standards. R is an open-source software environment with state-of-the-art statistical packages for the analysis of optical imagery. However, it lacks the tools for providing unified access to multi-program archives to customize and process the time series of images. This manuscript introduces `RGISTools`, a new software that solves these issues, and provides a working example on water mapping, which is a socially and environmentally relevant research field. The case study uses a digital elevation model and a rarely assessed combination of Landsat-8 and Sentinel-2 imagery to determine the water level of a reservoir in Northern Spain. The case study demonstrates how to acquire and process time series of surface reflectance data in an efficient manner. Our method achieves reasonably accurate results, with a root mean squared error of 0.90 m. Future improvements of the package involve the expansion of the workflow to cover the processing of radar images. This should counteract the limitation of the cloud coverage with multi-spectral images.

**Keywords:** Landsat; R software; satellite images; Sentinel; spatio-temporal data

## 1. Introduction

Satellite images represent a valuable data source in large-scale long-term research studies. Landsat, MODIS, and Copernicus are major programs for the acquisition of images of the Earth's surface supported by the U.S. Geological Survey (USGS), NASA, and the European Space Agency (ESA), respectively. Images are freely accessible in large data archives, which can be retrieved via web services such as EarthData, NASA Inventory, or SciHub. Data archives offer long series of records, dating back to 1972 for Landsat, 1999 for MODIS, and 2013 for Sentinel. Satellite imagery has proven useful for studies in many disciplines, such as economic assessments [1], hydrological studies [2], soil property quantification [3,4], the distribution of animal species [5], and agricultural monitoring [6].

Missions have strengths and weaknesses regarding the spatial resolution and temporal frequency of their imagery. The satellite constellation of MODIS acquires images on a daily basis at a moderate spatial resolution (250 m). Landsat and Sentinel multispectral constellations capture high-resolution images (15–60 m and 10–60 m, respectively), and locations are revisited roughly on a weekly basis (eight and five days, respectively). Studies claim the need for a higher spatio-temporal resolution than those obtained from single programs [7]. Data fusion has been proposed to counteract inadequate

resolutions by blending information at different levels, such as pixel (e.g., MODIS and Sentinel), feature (e.g., class of land cover), or the decision level [8]. Data fusion is a cost-effective solution that is increasingly popular thanks to improvements in the availability and accessibility of satellite images from several platforms. Yet, web services and programs work with particular query protocols, file formats, and data standards. Becoming familiar with the details of every archive can be tedious and time consuming. A harmonized single access point and processing software would benefit the research community by removing complexity and fostering data fusion.

R Core Team [9] is a free statistical software increasingly used for the analysis of satellite images, due to the many reliable packages to analyze spatial or spatio-temporal datasets (e.g., `raster` [10], `sf` [11], or `stars` [12]), implement state-of -the-art statistical techniques devoted to remotely sensed data (e.g., `dtSwat` [13], `RSToolbox` [14], or `waveformlidar` [15]), and visualize satellite images (e.g., `mapview` [16] or `tmap` [17]). Packages working with satellite images already exist in `R`, but none deal with imagery from several programs or cover the overall workflow with satellite images. For instance, `SkyWatchr` [18] finds and downloads Landsat, MODIS, Sentinel, and private company's imagery, but it does not support the processing or customization. Other packages have greater functionalities, but they are specialized in particular programs or data products. Specifically, the `MODIStsp` [19] package downloads, mosaics, re-projects, and computes spectral indices from MODIS images exclusively. The `sen2r` package [20] is able to find, download, and process data products only from Sentinel missions. Packages such as `landsat` [21] or `satellite` [22] perform radiometric and topographic corrections of Landsat imagery, but neither search nor download images from this program. Hence, there is a need for a comprehensive package that harmonizes the work with different satellite programs.

Here, we introduce the package `RGISTools` [23] (v1.0.2), as a toolbox to download, customize, and process time series of satellite images from Landsat, MODIS, and Sentinel in a standardized way. The download process includes searching and previewing the available images for a region and period of interest. The customization covers mosaicking, cropping, and extracting the required bands. Processing functions comprise cloud removal, index computation, gap filling, and outlier smoothing [24]. `RGISTools` is available from the Comprehensive `R` Archive Network and the GitHub repository.

Other stand-alone software alternatives are also available for users and practitioners. The software for the processing and interpretation of Remotely sensed image Time Series (SPIRITS) [25] not only facilitates the processing and analysis of large image time series for monitoring crop and natural vegetation, but also for drought detection among many other applications.

The aim of this paper is to provide an overview of the package through a case study where the water level of a reservoir is estimated. Freshwater bodies on the Earth's surface play a key role in natural ecosystems (e.g., water cycle and habitats) and the industrial sector (e.g., agriculture and electricity generation). Monitoring their variations is crucial, especially during extremes events such as flood or droughts, given their serious consequences on the economy and food-security [26]. Remote sensing is an effective and efficient source of information to constantly survey water bodies at multiple scales and for different purposes [27]. For example, it has been used to study the ecological impact of human activities on a lagoon in the Mediterranean [28], study the dynamics of large delta rivers [29], quantify flooded areas [30], or survey newly dammed water reservoirs [31].

The area, height, and volume of lakes are important hydrological parameters and ecological indicators [32]. Optical sensors are extensively applied to track lakes due to their high availability and wide range of spatio-temporal resolutions [2]. Estimating height and volume through optical imagery requires auxiliary information regarding lake bathymetry or altimetry. In situ measurements of the lake bathymetry are scarce [33,34], so digital elevation models (DEM) from satellite spectral or radar sensors, such as the Advanced Spaceborne Thermal Emission and Reflection Radiometer (ASTER) [35,36], the Shuttle Radar Topographic Mission (SRTM) [37,38], or TanDEM-X DEM [39,40], can be used as alternatives. However, DEMs from space are limited by their vertical accuracy and spatial resolution. Furthermore, in most cases, the method can only be used for newly filled reservoirs or

other water bodies where the water level has risen above its level at the time the elevation data were acquired, because most terrestrial DEMs do not represent the bathymetry within existing water bodies. Another option is measuring directly the water lamina height from active sensors, such as Multiple Altimeter Beam Experimental Lidar (MABEL) [41], ICEsat [42,43], and D-InSAR [44], among others. These methods show higher levels of accuracy, but data need more complex processing and less frequent revisiting times [45].

To illustrate the package contribution to this important endeavor, the case study estimates the level of water in a reservoir located in Northern Spain using optical multi-spectral imagery. The information on terrain elevation is approximated by means of a topographic map made by the local authorities before the construction of the dam.

The structure of the paper is as follows: Sections 2 and 3 introduce basic background information about the study site and the methodological approach, respectively. The latter section includes the R code and a brief explanation of the package functions. Section 4 presents the results of the analysis, and Section 5 discusses the utilities and limitations of the package. The major concluding remarks are summarized in Section 6.

## 2. Study Site

The Itoiz reservoir is strategically placed in the foothills of the Pyrenees with its dam wall northeast of the the village of Aoiz (Figure 1). The dam collects the water from the Irati River, extending over 1100 ha, and has a maximum capacity of 418 hm$^3$. The reservoir was built between 1996 and 2006 and became fully operational in 2008. It provides freshwater to nearly 354,500 people and irrigation water to 57,713 ha of agricultural fields.

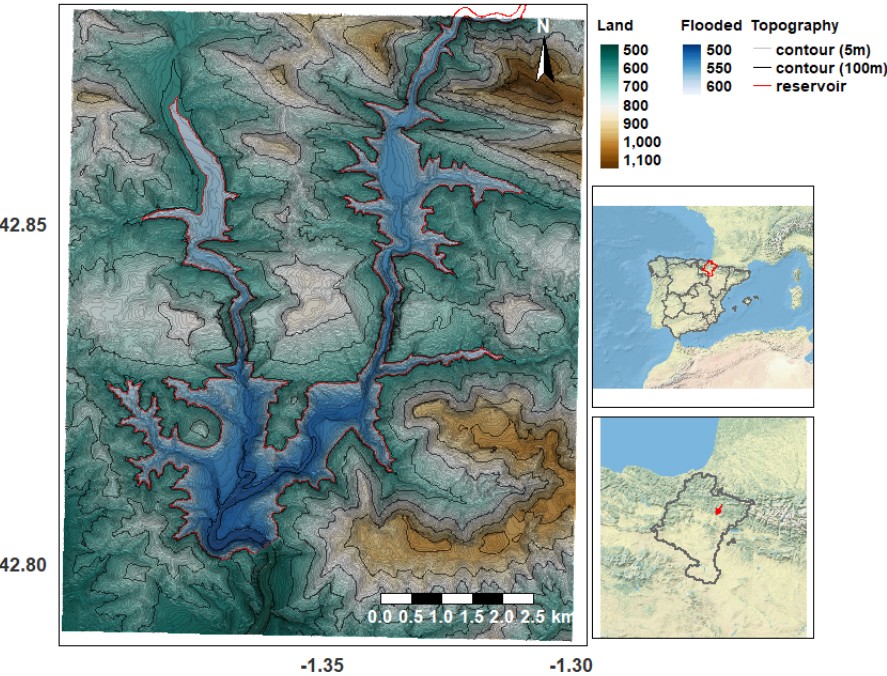

**Figure 1.** Elevation map of the study region, including the Itoiz reservoir basin. The map on the left represents the original contour map in grey, and black lines show five and 100 meter variations in altitude. The continuous color code displays the gridded altimetry interpolated at a $10 \times 10$ m$^2$ resolution from the topographic map. The maps on the right, from top to bottom, display the location of Navarre (red border) in Spain and the location of the reservoir (Itoiz) within Navarre.

## 3. Materials and Methods

The case study was an application of the package's main functionalities, such as retrieving, customizing, and processing time series of multi-spectral satellite images (Figure 2). The use-case combined Landsat-8 and Sentinel-2 imagery to monitor the water levels of a reservoir named Itoiz. The dam is located close to Pamplona, which is the main population of the Navarre province in Northern Spain. Few studies have explored this combination of satellite programs to monitor water bodies (except, e.g, [46,47]). Similar to [48], during the analysis (Figure 2), the water levels were obtained by superimposing the pond's shorelines detected through satellite imagery and a topographic map of the basin (digital elevation model (DEM) in Figure 2). The estimates were tested against measurements taken at the dam wall (Obs. in Figure 2).

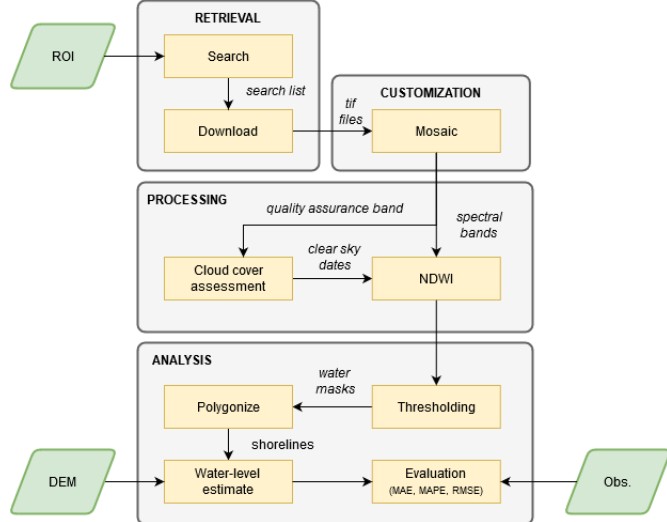

**Figure 2.** Workflow diagram of the approach in the case study of the Itoiz reservoir. The workflow is divided into four steps (grey zones); retrieval, customization, processing, and analysis. Steps 1–3 show the main capabilities of `RGISTools`. Step 4 illustrates other analytical tools `R`. Green squares represent the information inputs, where ROI stands for the region of interest, DEM for digital elevation model, and Obs. for in situ observations of the water levels. The evaluation was carried out using the mean absolute error (MAE), the mean absolute percentage error (MAPE), and the root mean squared error.

Section 3.1 introduces the auxiliary data required for the analysis (i.e., topographic map and water level measurements), and Section 3.2 explains in depth the implementation of the analysis.

*3.1. Materials*

3.1.1. Auxiliary Data

In `R`, the bounding box that encapsulates the reservoir can be defined as follows, using a latitude-longitude coordinate reference system:

```
> library("sf")
> roi.bbox <- st_bbox(c(xmin = -1.40, xmax = -1.30,
ymin = 42.79, ymax = 42.88), crs = 4326)
> roi.sf <- st_as_sf(st_as_sfc(roi.bbox))
```

The topographic map of the region is freely available at the website of the Spatial Data Bureau of Navarre (IDENA) [49]. The map was originally obtained applying photogrammetric restitution from aerial pictures captured during July 2000, i.e., before the dam was built. The map was used as a proxy of the elevation in the hillsides of the watershed, and it may not be sufficiently accurate to represent the elevation in the bed of the lake. Therefore, the analysis was restricted to the water lamina height.

The topographic map represented the elevation of the terrain as contours for every 5 m elevation variations (contour lines in Figure 1). Contour polygons were rasterized and interpolated at a resolution of $10 \times 10$ m$^2$, applying the inverse distance weighting (IDW) method. The resulting elevation map can be seen as gradient colors in Figure 1.

Water levels are measured on a daily basis at the wall in meters above sea level (m.a.s.l.). Measurements are publicly available on-demand at the Automatic Hydrological Information System (SAIH), managed by the Ebro River Basin Authority [50].

### 3.1.2. Satellite Imagery

We used Landsat-8 and Sentinel-2 Level-2 products in our analysis [51,52]. This meant that the images were atmospherically and topographically corrected, providing the bottom-of-atmosphere surface reflectance. Topographic corrections might be particularly important in this reservoir to rectify the shadows caused by the surrounding mountainous terrain. The Landsat-8 and Sentinel-2 tiles that covered the reservoir were 200-30 and T30TXN, respectively. The time series expanded over the period 1 July 2018 and 1 May 2019. This was the time of the year that the water storage varied the most, and therefore, it was more interesting to analyze.

In order to extract the entire water shoreline, images partially covered by clouds that obstructed the view of the reservoir were discarded. As a result, a total of 6 and 21 surface reflectance scenes from Landsat-8 and Sentinel-2 were used in the analysis. Sentinel-2 images were resampled from their 10 m pixel resolution to match the 30 m resolution of the Landsat-8 images. The surface reflectance was used to derive the normalized difference water index (NDWI). The NDWI is a remote sensing index usually applied to detect flooded areas [53]. It has been used extensively to map water bodies from multi-spectral satellite images [54]. The NDWI marks out water bodies based on the strong absorbability in the near-infra-red band (NIR) and the strong reflectance in the green band (green) from water (Equation (1)). Pixels with an NDWI above 0 are candidates for open water bodies, although thresholds between 0 and 0.1 are frequently adopted [55]. The equation to compute the NDWI is as follows:

$$NDWI = \frac{(Green - NIR)}{(Green + NIR)} \tag{1}$$

Other indices have been proposed in the literature to avoid the misclassification of built-up areas as water, such as the modified NDWI (MNDWI) [56]. We argue that the NDWI was a suitable index for illustration purposes in our case study due to its long and robust trajectory and the absence of built-up areas around the reservoir.

### 3.2. Methods

This section shows how to perform the data acquisition and processing using `RGISTools` (v1.0.2) and explains the methodological approach to estimate the water levels of the Itoiz reservoir.

### 3.2.1. Searching

The functions `lsSearch()` and `senSearch()` scan the Landsat and Sentinel-2 repositories to find the scenes that match the requested data product (`product`), time interval (`dates`), and ROI (`region = roi.sf`). The functions trace the satellite images in the SciHub [57] and EarthData [58] archives, respectively.

Landsat and Sentinel search functions allow filtering the results by cloud coverage with the `cloudCover()` function. We restricted our search to images with a cloud coverage between 0 and 80% (`cloudCover = c(0,80)`) to avoid likely obstructed views during winter. Since the tiles covered a larger area than the reservoir, a more restrictive cloud coverage may cause the loss of scenes with clear skies in our region of interest. Further assessments of cloudiness are conducted in Section 3.2.4.

Landsat only provides immediate access to Level-1 products (`product = "LANDSAT_8_C1"`). To obtain the Level-2 product, we must search first the Level-1 images and then request their correction from the Earth Resources Observation and Science (EROS) Center through their Science Processing Architecture (ESPA) [59] at the time of downloading:

```
> library("RGISTools")
> sres.ls8 <- lsSearch(product = "LANDSAT_8_C1",
dates = as.Date("2018-07-01") + seq(0, 304, 1),
region = roi.sf,
cloudCover = c(0,80),
username = "USERNAME",
password = "PASSWORD")
```

Regarding Sentinel-2, surface reflectance images are instantly available with the product "S2MSI2A" (Sentinel-2 MultiSpectral level-2A):

```
> sres.sn2 <- senSearch(platform = "Sentinel-2",
product = "S2MSI2A",
dates = as.Date("2018-07-01") + seq(0, 304, 1),
region = roi.sf,
cloudCover = c(0,80),
username = "USERNAME",
password = "PASSWORD")
```

Note that both `lsSearch()` and `senSearch()` require log-in credentials to access the EarthExplorer and SciHub repositories. Replace `USERNAME` and `PASSWORD` with the reader credentials after signing up for both web services in [60,61].

### 3.2.2. Downloading

The `lsDownload()` and `senDownload()` functions retrieve the time series of satellite images found in the previous section (`sres.ls8` and `sres.sn2`).

Landsat-8 images must be atmospherically corrected by EROS, so we set `lvl=2` in `lsDownload()`, which automatically makes a request to ESPA. The requests can be named using the `l2rqname` argument to distinguish the current request from others. With the `bFilter` argument, we can specify the bands needed, which for our purpose were the green ("band3") and near-infra-red ("band5") bands to compute the NDWI [53] and the quality ("pixel_qa") band to further analyze the cloud coverage. Images were then automatically saved as "GTiff" in the `./Landsat8/untar` directory:

```
> wdir.ls8 <- file.path(wdir, "Landsat8")
> lsDownload(searchres = sres.ls8,
lvl = 2,
untar = TRUE,
bFilter = list("band3", "band5", "pixel_qa"),
username = "USERNAME",
password = "PASSWORD",
l2rqname = "REQUESTNAME",
AppRoot = wdir)
```

Similarly, `senDownload()` downloads Sentinel imagery. In Sentinel-2, Bands 3 and 8 correspond to green a near-infra-red wavelengths. Both bands are available at a resolution of 10 m. Hence, we refer to them as "B03_10m" and "B08_10m". Information about the cloud coverage is in the cloud probability band. This band is provided at a maximum resolution of 20 m (i.e., "CLDPRB_20m"). The files are saved as "GTiff" in the `./Sentinel2/unzip` directory:

```
> wdir.sn2 <- file.path(wdir, "Sentinel2")
> senDownload(searchres = sres.sn2,
unzip = TRUE,
bFilter = list("B03_10m", "B08_10m", "CLDPRB_20m"),
username = "USERNAME",
password = "PASSWORD",
AppRoot = wdir.sn2)
```

### 3.2.3. Mosaicking and Cropping

Mosaicking means merging the satellite images with the same capturing date, but coming from different tiles to obtain a single scene covering the ROI. Cropping is the removal of pixels outside the spatial bounding box of the ROI. Both tasks are meant to rearrange the dataset and preserve the relevant information only. The functions that crop and mosaic Landsat and Sentinel images are `lsMosaic()` and `senMosaic()`. The functions take the scenes saved in ./Landsat8/untar and ./Sentinel2/unzip and place the results into two folders created automatically; ./Landsat8/ls8_itoiz and ./Sentinel2/sn2_itoiz):

```
> wdir.ls8.untar <- file.path(wdir.ls8, "untar")
> lsMosaic(src = wdir.ls8.untar,
region = roi.sf,
out.name = "ls8_itoiz",
gutils = TRUE,
AppRoot = wdir.ls8)

> wdir.sn2.unzip <- file.path(wdir.sn2, "unzip")
> senMosaic(src = wdir.sn2.unzip,
region = roi.sf,
out.name = "sn2_itoiz",
gutils = TRUE,
AppRoot = wdir.sn2)
```

### 3.2.4. Cloud Mask Filtering

The `lsCloudMask()` and `senCloudMask()` functions interpret the bits of the quality bands to generate clouds masks, i.e., images where there are 1s and NAs indicating clear-sky or cloudy pixels, respectively. Functions require the location of the quality bands, which must be passed through the `src` argument. The outputs are saved in the `AppRoot` directory, in a new folder named as `out.name`:

```
> wdir.ls8.mosaic <- file.path(wdir.ls8, "ls8_itoiz")
> lsCloudMask(src = wdir.ls8.mosaic,
out.name = "ls8_cldmask",
AppRoot = wdir.ls8)

> wdir.sn2.mosaic <- file.path(wdir.sn2, "sn2_itoiz")
> senCloudMask(src = wdir.sn2.mosaic,
out.name = "sn2_cldmask",
AppRoot = wdir.sn2)
```

We load the cloud masks into R to conduct further analyses of cloud coverage:

```
> wdir.ls8.cld <- file.path(wdir.ls8, "ls8_cldmask")
> wdir.sn2.cld <- file.path(wdir.sn2, "sn2_cldmask")
```

```
> wdir.all.cld <- list(wdir.ls8.cld, wdir.sn2.cld)
> fils.cld.msk <- lapply(wdir.all.cld, list.files, full.names = TRUE)
> imgs.cld.msk <- lapply(files.cld.msk, stack)
> names(imgs.cld.msk) <- c("ls8", "sn2")
```

The cloud coverage ratio is calculated as the fraction of the number of NAs and the overall number of pixels in the image:

```
> cld.coverage <- lapply(imgs.cld.msk,
function(x){
colSums(is.na(getValues(x)))/ncell(x)
})
> names(cld.coverage) <- c("ls8", "sn2")
```

Clear-sky images are considered those whose fraction of NAs remains below a given threshold. The threshold was set to 20% for Landsat-8 and 0.1% for Sentinel-2 images. These thresholds were decided through visual inspection and comparisons between scenes and cloud masks. Landsat-8 has a higher threshold than Sentinel-2 due to misclassified shadows as cloudy pixels:

```
> ls8.clr.imgs <- which(cld.coverage$ls8 < 0.20)
> sn2.clr.imgs <- which(cld.coverage$sn2 < 0.001)
> ls8.clr.dates <- genGetDates(names(imgs.cld.msk$ls8))[ls8.clr.imgs]
> sn2.clr.dates <- genGetDates(names(imgs.cld.msk$sn2))[sn2.clr.imgs]
```

Both `ls8.clr.dates` and `sn2.clr.dates` represent the dates with clear skies at the reservoir.

### 3.2.5. Deriving the Ndwi

`RGISTools` provides several built-in functions to compute remote sensing indices. Among them, there is the NDWI (`varNDWI()`). Both `ls8FolderToVar()` and `senFolderToVar()` apply `varNDWI()` to the time series of images considered so far. The functions `FolderToVar` are responsible for matching the band names in `varNDWI()` with the appropriate band numbers in each mission. The NDWI is only computed for those dates with clear-sky images as follows:

```
> wdir.ls8.mosaic <- file.path(wdir.ls8, "ls8_itoiz")
> ls8FolderToVar(src = wdir.ls8.mosaic,
fun = varNDWI,
dates = ls8.clr.dates,
AppRoot = wdir.ls8)

> wdir.sn2.mosaic <- file.path(wdir.sn2, "sn2_itoiz")
> senFolderToVar(src = wdir.sn2.mosaic,
fun = varNDWI,
dates = sn2.clr.dates,
AppRoot = wdir.sn2)
```

The time series of NDWIs from the Landsat-8 and Sentinel-2 imagery are automatically saved at `./Landsat8/NDWI` and `./Sentinel2/NDWI`, respectively.

### 3.2.6. Detecting Water and Level Analysis

The NDWI images can be loaded in R as follows:

```
> imgs.ndwi <- list(
stack(list.files(file.path(wdir.ls8,"NDWI"), full.names = TRUE)),
stack(list.files(file.path(wdir.sn2,"NDWI"), full.names = TRUE)))
```

Layers receive the name of the index and their capturing date (e.g., "NDWI_2018244"). To keep track of the source of every image, we additionally paste a mission label ("LS8" and "SN2") to the names of the layers:

```
> names(imgs.ndwi[[1]]) <- paste0(names(imgs.ndwi[[1]]), "_LS8")
> names(imgs.ndwi[[2]]) <- paste0(gsub("10m", "SN2", names(imgs.ndwi[[2]])))
```

In the following code, the Sentinel-2 series is reprojected and resampled in order to match the resolution and projection of the Landsat-8 imagery so they can be combined into a single object. The resampling is carried out due to functional and technical reasons. As a single object, the time series is easier to handle and the code shorter, more readable, and faster to run. Resampling also smooths noisy pixels, which may cause problems in subsequent steps of the analysis. Furthermore, the layers are re-arranged to follow the temporal sequence, regardless of their mission:

```
> imgs.ndwi[[2]] <- projectRaster(imgs.ndwi[[2]], imgs.ndwi[[1]])
> imgs.ndwi <- stack(imgs.ndwi)
> imgs.ndwi <- imgs.ndwi[[order(names(imgs.ndwi))]]
```

For every image, the NDWI is translated into water levels as follows:

1.  Detection of flooded pixels: Pixels above the thresholds $-0.16$ (Landsat-8) and $-0.1$ (Sentinel-2) are considered as flooded in the NDWI series. The thresholds were selected by visual inspection.
2.  Shoreline identification: This aims at joining the flooded pixels into a single entity. The individual flooded pixels are transformed into polygons, a class object in R. This class allows removing the interior boundaries among them, merging all the flooded pixels into a single polygon. Isolated misclassified pixels or scattered polygons are dismissed. The edge of the largest polygon defines the main water body shoreline.
3.  Estimating water levels: We extract the shoreline elevations by intersecting the polygon with the topographic map. Elevations may contain some errors due to inaccuracies in the topographic map or the shoreline delineation. Hence, the elevations are used to derive a probability density distribution. The most likely elevation is the one with the highest density value.

```
> shorelns <- lapply(as.list(imgs.ndwi),
function(r){
thrsh <- ifelse(grepl("LS8",names(r)), -0.16, -0.10)
water <- rasterToPolygons(clump(r> thrsh), dissolve = TRUE)
shors <- st_union(st_as_sfc(water))
bodis <- st_cast(shors, "POLYGON")
areas <- st_area(bodis)
st_sf(st_cast(bodis[which.max(areas)], "MULTILINESTRING"))
}
)
> shorelns.z <- lapply(shorelns,
function(pol, altimetry.itoiz){
line <- as(as(pol, "Spatial"), "SpatialLines")
cell <- cellFromLine(altimetry.itoiz, line)[[1]]
elvs <- getValues(altimetry.itoiz)[cell]
dnst <- density(elvs, kernel = "epanechnikov", na.rm = T)
dnst$x[which.max(dnst$y)]
}, altimetry.itoiz)
> level.est <- unlist(shorelns.z)
```

The R scripts with the code and datasets can be found as Supplementary Material in the GitHub repository [62]. An additional script is included for estimating the water volume, the length of the shoreline, and the water surface of the reservoir, through the lakemorpho library [63].

### 3.2.7. Evaluation

Estimates were compared with in situ measurements of the water levels through the mean absolute error (MAE, Equation 2), the mean absolute percentage error (MAPE, Equation 3), and the root mean squared error (RMSE, Equation 4), defined as follows:

$$MAE = \frac{\sum_{i=1}^{n} | \hat{y}_i - y_i |}{n} \tag{2}$$

$$MAPE = \frac{1}{n} \sum_{i=1}^{n} \frac{| \hat{y}_i - y_i |}{y_i} * 100 \tag{3}$$

$$RMSE = \sqrt{\frac{\sum_{i=1}^{n} (\hat{y}_i - y_i)^2}{n}}, \tag{4}$$

where $y_i$ and $\hat{y}_i$ are the $i^{th}$ observed and estimated water levels, respectively, from $i = 1, \ldots, n$. The MAE and RMSE provide an overall measurement of joint agreement. The MAPE provides a relative measurement of the error magnitude.

Additionally, we assessed the performance of `RGISTools` (v1.0.2) by measuring the time and memory consumption of the analysis. The experiment was conducted in a computer with an Intel(R) Core(TM) i7-6700 CPU @3.40 GHz processor and an internet connection speed of 310 Mbps.

## 4. Results

### 4.1. Flooded Area and Level Estimates

We captured 28 images between 1 July 2018 and 1 May 2019. Of those, seven images corresponded to Landsat-8 scenes and 21 to Sentinel-2 images. Images covered the entire period of analysis, with the only exception of February 2019. This month showed higher levels of precipitation, which was linked to a greater cloudiness and a rapid rise of the water levels in the reservoir (Figure 3).

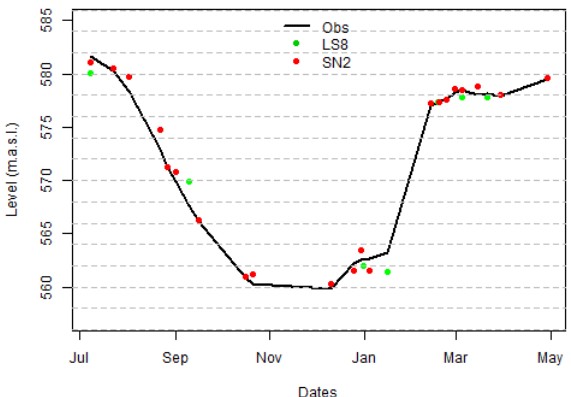

**Figure 3.** Water levels in the Itoiz reservoir between 1 July 2018 and 1 May 2019. The water levels are in meters above sea level (m.a.s.l.). The black line represents the observations (Obs). Green and red dots are estimates from Landsat-8 (LS8) and Sentinel-2 (SN2), respectively. The dashed line (Est) shows the combination of Landsat-8 and Sentinel-2 water levels.

The NDWI images (Figure 4) showed a good separability between water and vegetation. Flooded areas corresponded to positive values of the NDWI (blue color) and the surroundings with negative values of the index (brown color). Images revealed that some aspects could potentially interfere with the proper identification of shorelines. In spite of the preprocessing, Landsat-8 imagery exhibited shadows during winter in the hillsides oriented northeast. Furthermore, some clouds persisted in the final collection of images (e.g., "NDWI_2018189_SN2" or "NDWI_2018253_LS8" in Figure 4). However, they did not obstruct the view of the main water body of the reservoir.

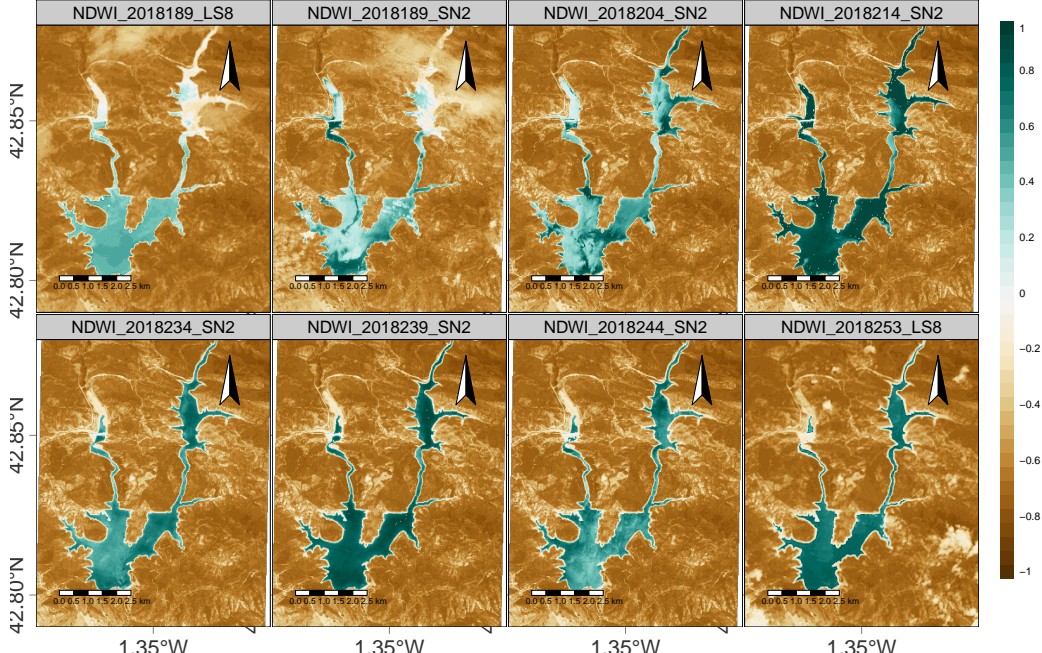

**Figure 4.** Water detection (blue color) at the Itoiz reservoir. The first eight instances of the NDWI time series from Landsat-8 ("LS8") and Sentinel-2 ("SN2"). The "x" and "y" axes are the longitude and latitude coordinates. The names of the panels additionally show the capturing date of the image in `YYYYJJJ` format, where `Y` and `J` represent a year and a Julian date digit, respectively.

Estimates represented well the variations of the water levels measured in situ. The MAE showed a good agreement between the ground truth data and the estimates (0.66 m) for both satellites combined. Landsat-8 imagery led to a larger MAE (1.04 m) than Sentinel-2 (0.53 m), which represented 0.18% and 0.09% in relative terms. Regarding the RMSE, the performance of the estimates from the two satellites combined was 0.90 m. As with the MAE, the RMSE of Landsat-8 estimates was larger (1.29 m) than the estimates from Sentinel-2 (0.73 m).

Rgistools (V1.0.2) Performance

We evaluated the time that it took less to download, customize, and process the 28 images involved in the analysis. Most of the time was invested in downloading the satellite images (Section 3.2.2), partly due to the on-demand corrections applied by ESPA on the Landsat-8 imagery. The next most time-consuming process was the cropping and mosaicking (Section 3.2.3), taking 0.25 h to run. `R` is an interpreted language, and so, the processing speed is slower than other computer languages, such as `C`. Therefore, `RGISTools` relied on the Geo-spatial Data Abstraction Library [64] for some steps of the workflow, such as format transformations, cropping, or mosaicking.

The analysis of the reservoir required 81.24 GB of memory disk. Most memory was needed during the download and decompression phase (Section 3.2.2). Due to the small extension of the region of interest, the size of the image collection decreased from 81.24 GB to 0.25 GB after cropping (Section 3.2.3). At this point, the original images could be removed, freeing disk space. Note that satellite images (cloud masks and NDWI imagery) were loaded in `R` near the end of the processing (cloud coverage and the water lamina were analyzed), when the size of the scenes summed in total 0.07 GB (Section 3.2.6).

## 5. Discussion

The case study used optical satellite imagery to estimate the water level of a reservoir. Estimates were based on the NDWI index and auxiliary data of terrain elevation. There are multiple

remote sensing indices devoted to water detection with pros and cons that should be considered before doing the analysis [65]. The NDWI shows good performance in particular occasions [66], but mixes the spectral signature from water bodies and built-up areas [56]. In our case study, the surrounding area of the lake was dominated by wild vegetation, favoring a good separation between flooded and non-flooded pixels. We took a closer look at an urban built-up area of the study region (see Appendix A) to ensure the proper operation of the NDWI. Visual inspection revealed that a few scattered pixels from the built-up area were wrongly classified as flooded (Figures A2 and A3). As they were separate from the main water body, Step 2 of the analysis (Section 3.2.6) removed the pixels before retrieving their elevation in Step 3. Furthermore, the analysis considered the most likely elevation value for the final estimate so they were robust to small localized errors.

Regarding the accuracy of the water level estimates, other studies reported similar levels of performance. For instance, the work in [48] had slightly smaller errors than those obtained for the Itoiz reservoir ($RMSE = 0.76$ m and $MAE = 0.601$ vs. $RMSE = 0.90$ m and $MAE = 0.61$), when using a similar, but more convoluted technique for Lake McConaughy (United States) using 30 m resolution Landsat-5 imagery. The work in [45] reported RMSEs ranging from 0.85 to 1.90 m for the Hoover Dam in Lake Mead (United States) using Landsat imagery and different digital elevation models. Although good for reference, comparisons with other studies should be taken with care. Despite the lack of evidence, the work in [47] suggested that errors were case-specific since lake morphology could theoretically affect the accuracy of the results. In this regard, Lake McConaughy and Hoover Dam extend over larger areas than the Itoiz reservoir (90 and 280 km$^2$ vs. 11 km$^2$). The assessment of [47] involved different morphologies, but it was conducted over a longer time period.

As in [48], the thresholds were manually fixed for each satellite program based on the ability of separating water from land. The threshold disparity between satellite programs might be justified by differences in the satellite instruments and correction algorithms [67]. The code provided could be easily adapted to incorporate the application of dynamic threshold techniques. One of the most popular, the Otsu technique [68], separates the pixel values into two groups by maximizing the variances between them. The application of this technique in R would be straightforward by using other packages, such as `autothresholdr` [69]. Our case study did not apply this technique since the dataset did not meet the requirements to find the threshold automatically. Due to the shape of the lake, the flooded area represented a small portion of the overall image (below 20%). Since land dominated the image, NDWI histograms were unbalanced, and the variances within the two groups differed. Then, Otsu returns a biased threshold towards the group with the largest variance [70], leading to error in the water level estimates. Other refinements such as the normalization of the surface reflectance could help with this issue. However, it should be noted that the case study was an illustration of a newly developed package. Only the key processing steps were applied in order to achieve a scientifically accurate while reproducible example.

Cloud coverage is a major constraint when monitoring water bodies with multi-spectral satellite images. From the overall 19 and 49 images available from the Landsat-8 and Sentinel-2 (A and B) satellites for the period of analysis, only five and 21 were finally suitable for the proper identification of the reservoir shoreline. Blending optical images with synthetic aperture radar (SAR) shows promising results for filling data gaps [71] and improves the temporal resolution. Currently, `RGISTools` can search and download SAR images from Sentinel-1, but it does not provide tools for processing. Future versions of the package are expected to include this kind of functionality.

Regarding the package, it worked locally with the time series of satellite images and could be challenging for RAM and disk memories. The package used three strategies to address these challenges. First, it applied efficient routines such as those in the Geo-spatial Data Abstraction Library [64] whenever possible. Second, it allowed removing or filtering unnecessary information for specific purposes through functions and arguments. Images were loaded in R at the end of the process, when images contained only essential information for a specific task.

We argue that working locally with satellite images is a sensible option for statisticians and environmentalists that pursue the development of new methods. `R` is a flexible environment to test tentative methods rapidly. Furthermore, the eager evaluation of the code enables immediate assessments of the results. Finally, `R` is an open source programming language that favors a better understanding, application, and enhancement of existing spatio-temporal methods.

## 6. Conclusions

Satellite images are valuable sources of information for a variety of applications. Monitoring water bodies is particularly relevant due to the fundamental role that lakes and ponds play in agriculture and monitoring vegetation. Multi-spectral satellite images are commonly used for this endeavor due to the availability and accessibility to longstanding archives. Several sources of satellite images exist in a wide range of formats and spatio-temporal resolutions, and their combination has been shown to bring benefits in assessing Earth's surface dynamics.

`R` is an open access software suitable for manipulating, analyzing, and visualizing satellite imagery, yet it lacks a comprehensive tool to retrieve and process time series from multiple platforms in a homogeneous manner. `RGISTools` (v1.0.2) was presented here through a case study focused on water body monitoring, yet it could be used in many other agricultural and environmental applications.

The analysis was carried out on a reservoir in Northern Spain as an opportunity to test the package and the use of Landsat-8 and Sentinel-2 imagery for water monitoring against ground truth data. The package demonstrated its ability to download, customize, and process series of images effectively and efficiently from both programs. Additionally, the case study showed how multi-spectral information could characterize reasonably well the variations of water levels (RMSE 0.90 m) with the aid of a topographic map. It also provided the `R` script for calculating the water volume of the reservoir [62] using the `lakemorpho` library [63]. Future improvements involve the extension of the customization and processing workflow to radar data to counteract the limitations triggered by cloud coverage on multi-spectral imagery.

**Author Contributions:** Conceptualization, U.P.-G., M.M.-S., A.F.M., and M.D.U.; methodology, U.P.-G. and M.M.-S.; software, U.P.-G. and M.M.-S.; formal analysis, U.P.-G., M.M.-S., and A.F.M.; writing, original draft preparation, M.M.-S.; writing, review and editing, U.P.-G., A.F.M., and M.D.U.; supervision, A.F.M. and M.D.U. All authors have read and agreed to the published version of the manuscript.

**Funding:** This research was supported by the project MTM2017-82553-R(AEI/FEDER, UE). It has also received funding from the la Caixa Foundation (ID1000010434), the Caja Navarra Foundation, and and UNED Pamplona, under Agreement LCF/PR/PR15/51100007.

**Acknowledgments:** The authors would like to thank both the Editor and the three referees for the constructive comments that led to the great improvement of this paper.

**Conflicts of Interest:** The authors declare no conflict of interest.

## Abbreviations

The following abbreviations are used in this manuscript:

| | |
|---|---|
| ASTER | Advanced Spaceborne Thermal Emission and Reflection Radiometer |
| CPU | Central processing unit |
| DEM | Digital elevation model |
| EROS | USGS Earth Resources Observation and Science |
| ESPA | EROS Center Science Processing Architecture |
| ESA | European Space Agency |
| GHz | Gigahertz |
| IDENA | Infraestructura de Datos Espaciales de Navarra |
| IDW | Inverse distance weighting |
| MABEL | Multiple Altimeter Beam Experimental Lidar |
| MAE | Mean absolute error |

| MAPE | Mean absolute percentage error |
| Mbps | Megabit per second |
| MNDWI | Modified normalized difference water index |
| MODIS | Moderate Resolution Imaging Spectroradiometer |
| NDWI | Normalized difference water index |
| NIR | Near infra-red |
| RMSE | Root mean squared error |
| ROI | Region of interest |
| SAIH | Sistema Automático de Información Hidrológica |
| SAR | Synthetic aperture radar |
| STRM | Shuttle Radar Topographic Mission |
| USGS | Unites States Geological Survey |

## Appendix A. Inspection of a Built-Up Area

Here, we inspect Nagore (Figure A1), the only urban built-up area by the lake, which is located at the northwest of the study region. In the following, we explore the situation in Nagore to gain further insights.

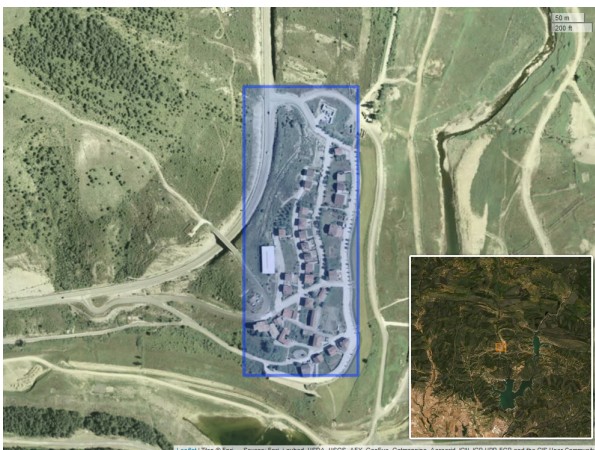

**Figure A1.** Satellite view of Nagore, a small community by the shoreline of the Itoiz reservoir. The main picture shows the buildings close to the shoreline. The smaller view at the bottom-right shows the location of Nagore (orange square) in relation to the lake. Images belong to the ESRI Wold Imagery Collection [72]. The graph was created using the package `leaflet` in R [73].

Figure A2 allows a visual inspection of the NDWI in the vicinity of Nagore. The figure shows the full series of images used in the analysis. Brown and blue colors correspond to values below and above zero, respectively. The blue polygon delimits the location of the buildings (the same polygon as in Figure A1). The dominant color within the rectangle is brown, corresponding to negative values of the NDWI. Water pixels tend to be positive, and they are shown in blue. The land area can be clearly distinguished from the lake, represented in white-blue tones. Figure A2 suggests a clear distinction between the built-up area and the lake.

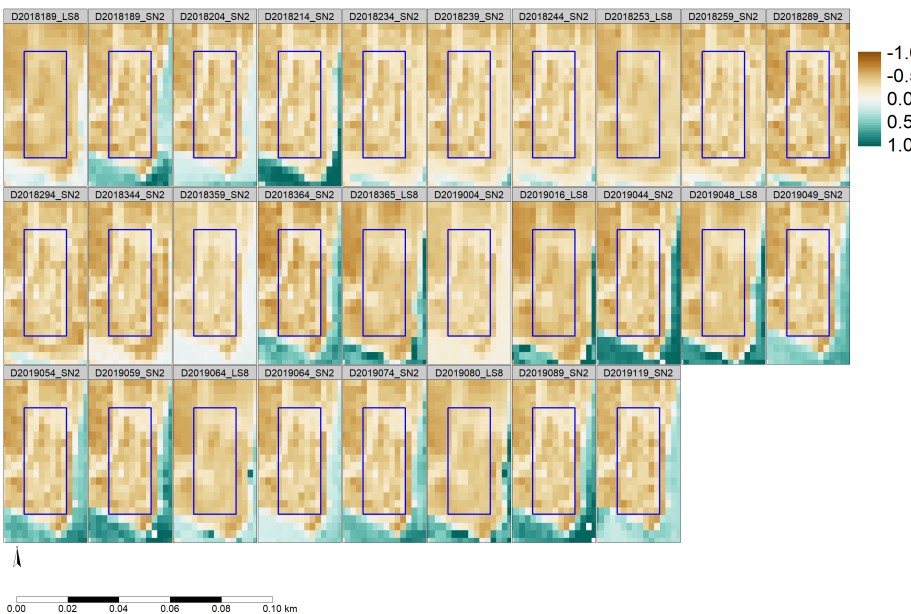

**Figure A2.** Classification of the NDWI series after thresholding. Figure labels show the capturing date of the satellite image using the year and Julian day (`YYYYJJJ`) format and an abbreviation of the satellite program (`LS8` and `SN2` stand for Landsat-8 and Sentinel-2.)

Figure A3 shows the same region as Figure A2, but the pixels are classified into water (blue) and non-water (white) based on two separate thresholds for Landsat-8 and Sentinel-2 ($-0.16$ and $-0.10$, respectively). The series shows that there are very few and scattered blue pixels within the rectangle. These pixels are separate from the water body (grouped blue pixels south and west Nagore), so they are discarded after the first filter of the algorithm. As a result, these exceptions are not considered for the elevation estimates.

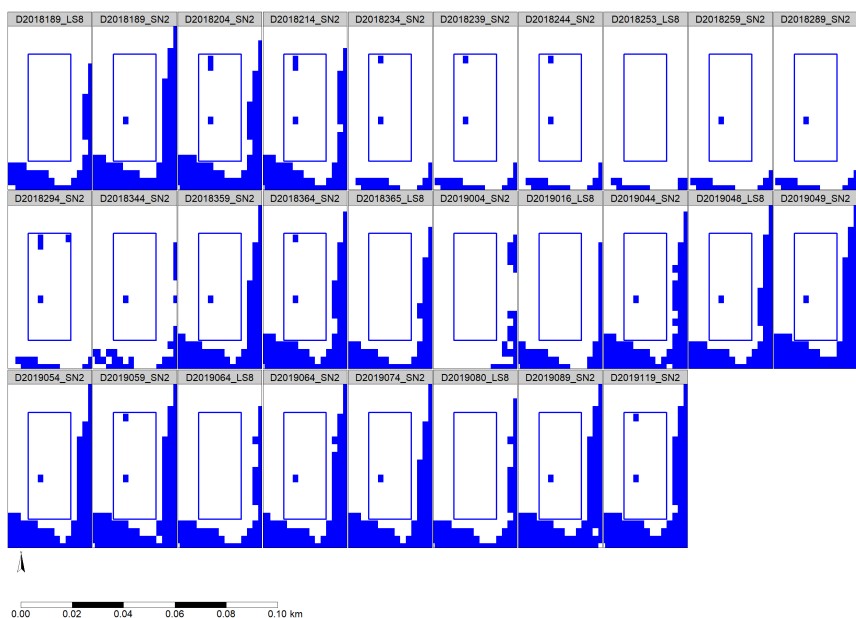

**Figure A3.** Time series of NDWI values in the vicinity of Nagore. Figure labels show the capturing date of the satellite image using the year and Julian day (`YYYYJJJ`) format and an abbreviation of the satellite program (`LS8` and `SN2` stand for Landsat-8 and Sentinel-2).

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
