# Peer review of "Using RGISTools to Estimate Water Levels in Reservoirs and Lakes"

_remotesensing, doi:10.3390/rs12121934_

Round 1
Reviewer 1 Report
The paper presents an R package that can comprehensively handle multi-spectral imagery from various sources. The authors also present a case study to estimate water level in a reservoir in Northern Spain. However, all the the methods presented in the package, although comprehensive, have been addressed before. The novelty in the paper is not obvious. The paper might be more suitable for a modeling/software journal, as opposed to remote sensing, since the paper does not present any novelty in research related to remote sensing.
Minor comments:
Please include line numbers in the manuscript when it is resubmitted.
Please cite relevant literature when applications in remote sensing are mentioned.
PDF with minor comments included.

Author Response
Please, find attached the file of the reply to your comments.

Reviewer 2 Report
This manuscript describes a new package of image processing tools in R, for the analysis of optical remote sensing images from Landsat and Sentinel-2. The package is introduced, and its use is illustrated by an example involving the remote measurement of water levels in a reservoir in Spain.
The paper is clear, well-written, and easy to understand. The toolkit it describes should be of interest to many readers; in fact, I would like to try it out myself. I have no major criticisms of the paper, but do have a number of suggestions for improvement, including some which are small but important.
[1] Abstract: First sentence is much too vague and general; it applies to most papers published in this journal. Also, the first five sentences of the abstract are all "introductory". I would recommend no more than two "introductory" sentences in the abstract before moving on to sentences about the data & methods.
[2] Abstract: "0.99 correlation" - see notes below re: autocorrelation
[3] Introduction, paragraph 1: references 1-5 appear to be more or less randomly chosen to represent the range of applications of analysis of satellite imagery. Several of the examples are decidedly narrow and obscure, but are cited as representative of much broader topics (e.g., a paper about the geographic distribution of Southern Hairy-Nosed Wombats represents all use of remote sensing in wildlife research). This is not particularly useful to most readers. If the authors want to demonstrate the breadth of the field, I would recommend choosing examples of papers that provide broader syntheses, e.g. review articles or others that are more summative rather than merely illustrative.
[4] Introduction: It is probably too late to be making this suggestion, but if the package is truly intended to be a "toolbox to download, customize, and process time-series of satellite images from Landsat, MODIS, and Sentinel", the name "RGISTools" is probably a bit too broad. Much GIS analysis does not involve use of satellite imagery at all, so it's a bit confusing to have a toolset of "GIS Tools" exclusively focused on this topic. That said, I can understand that it may be too late to change the name of the package.
[5] Data and Methods: On p. 3, just below Figure 1, the authors write: "Similar to [31], during the analysis (Fig. 1), the water levels are obtained by superimposing the pond’s shorelines detected through satellite imagery and a topographic map of the basin (digital elevation model or DEM in Fig. 1)" Variations on this approach have been described in the literature, beyond the single reference cited here. A quick search in Google scholar for the words "remote sensing lakes water level dem" brings up many references, some I am familiar with and some new ones. It would be worth saying a bit more about this method, since it is pretty much central to the paper. I would recommend adding the following:
(a) Cite more references to this method.
(b) Note that it requires having access to bathymetry (gridded elevation or depth data within the water body), which is often not available for many lakes worldwide.
(c) Note alternative methods, e.g. the use of altimetry (radar or laser altimetry, e.g. from ICESat-1 and -2) to measure water levels in lakes.
This could remain in this part of the manuscript, or move into the Introduction (literature review) if it seems too wordy for the methods section.
[6] Section 2.1.2 and Figure 2: As I understand it, the bathymetry of the reservoir shown in figure 2 was obtained by interpolation between vector contour lines on a topographic map. It would help to show the original map, and there is room on the page for a second panel in this figure (left, original contour map; right, gridded bathymetry interpolated from contours within the reservoir).
Often, such maps only show contours of elevation on land, not in lakes. The authors are fortunate if the original map included bathymetric contours, but that is often not the case. The need for a bathymetric digital elevation model, and their relative scarcity, should be mentioned by the authors somewhere in the discussion of methods and data.
[7] Section 2.1.3: Why resample the Sentinel-2 images to 30 m? Yes, the authors say it is to match the Landsat resolution, but why is that important? Is it necessary for both sensors used in the analysis to have the same resolution? That is a substantial loss of information, and it should be justified.
[8] Section 2.2.6: In step 2, the statement "The boundaries of neighboring cells are resolved" is unclear to me. Perhaps the authors are referring to what in GIS terminology is often labelled "dissolving"? "Resolved" to me in a remote sensing context suggests that something can be observed in the data (e.g., spatial, spectral, and temporal resolution).
[9] Overall, I like the approach (thresholding the water index into land/water, identifying the largest polygon, selecting the elevation grid cells along the land/water boundary, and determining the maximum likelihood value). As noted above, this has been used many times before, with slight variations on the methods.
However, I still haven't seen any reason in the methods why it is necessary to resample the Sentinel imagery to 30 meters. None of the steps described seem to necessitate all the imagery having the same resolution. Of course, having two different resolutions might affect the evaluation/validation step later (the higher resolution data might have a smaller standard error, or something) but that would also be interesting to see -- why not show it?
[10] I commend the authors for including the code and data in a GitHub repository. That's really helpful.
[11] Section 2.2.7: Without yet having seen the results, I am already concerned about the effect of temporal autocorrelation on the statistics used for evaluation. If the two time series (in-situ elevation measurements and remote sensing) are temporally autocorrelated, the measured correlation *between* them will be artificially "inflated" due to that autocorrelation. This will be less of a problem if measurements are being made far enough apart in time that autocorrelation is not a problem, but my guess is that the temporal spacing of images will be close enough that autocorrelation is important.
[12 ... immediate followup to previous ...] Results: Section 3.1: yes, I am very concerned that the reported "0.99" correlation coefficient is more indicative of autocorrelation than of the actual underlying robustnesss of the method. If the authors insist on reporting correlation values, they should at least discuss this issue clearly, ideally with a reminder each time they repeat the correlation value.
[13] Figure 4: I am not a fan of the dashed line connecting the remotely sensed data points. It implies that there is zero error in the points and that the underlying time series passes exactly through every point. It would be better to fit some model through these points, allowing for slight departures where points are close together in time (e.g., January). Since the authors are using R, they should have many options (e.g., a LOWESS smoothing function or something like that).
[14] Discussion: The paragraph beginning "As in [31], the thresholds were manually fixed..." is a bit confusing to me. Specifically, the sentence "Our case study does not apply this technique since the data-set does not follow the bi-modal frequency distribution that is required..." is very surprising. Are the authors saying that there is not a bi-modal distribution of values in the NDWI layer, with the modes representing water vs land? The water area should follow a fairly distinct and uniform distribution in NDWI. The land areas may be more complex, being the integration of different surface types with different individual distributions for NDWI values, but overall in my experience land areas should be distinctly different in their distribution of NDWI values from water, unless there is extensive areas with intermediate values e.g. wetlands or something. The authors should explain this. It might also be helpful to include a figure with superimposed histograms of NDWI values for the areas classified as land vs water.
[15] I am also a bit surprised that nowhere in the manuscript do the authors mention *volume* ... which can easily be derived from the data they have (the bathymetry and the water level). It's perfectly acceptable to keep the focus on water level, but it would make sense to at least discuss the obvious extension to calculate lake or reservoir volume.
Author Response
Please, find attached the reply of your comments.

Reviewer 3 Report
The paper entitled “Using RGISTools to estimate water levels in reservoirs and lakes” is a well written and relatively well structured paper of a very interesting topic. The paper is written in good English (I am not a native speaker but I understand that the manuscript is well written). The most important thing is the validation of the results derived from the remotely sensed methodology applied by the authors.
I have a couple of comments and suggestions that could improve the quality of the final version of the manuscript:
Section 2.1.1 Region of Interest should not be part of the Materials and Methods. It should be a separate section after the Introduction. This section should have a Figure showing the location of the study area within Spain and a more detailed map of the lake and the relief – topography of the broader area.
In my opinion all the step by step descriptions of the procedures given in section 2 should be moved at the end of the paper as an appendix. It is difficult for the reader to follow the text with all these “steps”.
Author Response
Please, find attached the reply to your comments.

Round 2
Reviewer 2 Report
The revisions to this manuscript (and the authors' response letter) have addressed all of my concerns with the previous version - except for one easily fixed issue (see below).
These changes have significantly improved the manuscript. I appreciate the authors' detailed response to my question about the bimodal distribution of NDWI values and agree that their response clarifies this. The new Appendix A is a helpful addition (I assume one of the other reviewers must have asked for this).
I have only three suggestions, two of which are simply typos:
[1] Lines 73-74: This entire new paragraph is very helpful, but perhaps I was not clear enough, because it still misses one big problem with using DEMs to measure water levels in reservoirs. The problem is that it usually only works when the water body is rising to a higher level than it was at the time the DEM was produced, so that the new shoreline falls on grid cells in the DEM that were dry land originally. Where the water level is falling rather than rising, most widely available DEMs don't provide bathymetric data within water bodies.
In other words, if the DEM includes a reservoir whose water level was at 140 m when the elevation data were collected, your method will work correctly for measuring water levels any time the reservoir rises above 140 m, but will not work if the water level drops below 140 m, because all the DEM grid cells along the new (lower) shoreline will just have values of 140 m.
This is not a problem for the case study here, because the elevation data were acquired before the reservoir was filled. But it does limit its applicability elsewhere.
I would suggest adding a second sentence in lines 73-75, to be something like this: "However DEMs from space are limited by their vertical accuracy and spatial resolution. Also, in most cases the method can only be used for newly filled reservoirs or other water bodies where the water level has risen above its level at the time the elevation data were acquired, because most terrestrial DEMs do not represent the bathymetry within existing water bodies."
Something like that.
[2] Line 75: Note spelling of Altimeter (not Aaltimeter)
[3] Line 474 (acronyms): Note spelling of MAPE ("Mean" Absolute ...)
Author Response
Your comment has been introduced in the revised version of the paper, and the typo has been repaired.
Thank you very much for your help.
We appreciate your positive revision of the paper.
Reviewer 3 Report
I suggest to the editor to accept the paper.
Author Response
Thank you for your comment.